# Biodegradation Mechanism of Polystyrene by Mealworms (*Tenebrio molitor*) and Nutrients Influencing Their Growth

**DOI:** 10.3390/polym16121632

**Published:** 2024-06-09

**Authors:** Hisayuki Nakatani, Yuto Yamaura, Yuma Mizuno, Suguru Motokucho, Anh Thi Ngoc Dao, Hiroyuki Nakahara

**Affiliations:** 1Graduate School of Integrated Science and Technology Chemistry and Materials Engineering Program, Nagasaki University, 1-14 Bunkyo-machi, Nagasaki 852-8521, Japan; bb52123639@ms.nagasaki-u.ac.jp (Y.Y.); bb54124140@ms.nagasaki-u.ac.jp (Y.M.); motoku@nagasaki-u.ac.jp (S.M.); anh.dao@nagasaki-u.ac.jp (A.T.N.D.); 2Organization for Marine Science and Technology, Nagasaki University, 1-14 Bunkyo-machi, Nagasaki 852-8521, Japan; 3Graduate School of Integrated Science and Technology Smart City Design Engineering Program, Nagasaki University, 1-14 Bunkyo-machi, Nagasaki 852-8521, Japan; nakaharahiroyuki@nagasaki-u.ac.jp

**Keywords:** mealworm, expanded polystyrene, biodegradation mechanism, benzene ring alternation, nutrients, growth

## Abstract

A degradation mechanism of polystyrene (PS) in mealworms reared on expanded PS (EPS) was investigated by its decrease in molecular weight and change in chemical structure. A 33% decrease in molecular weight was observed for the digested PS in the frass after 1 week of feeding to mealworms. The FT-IR and py-GC/MS spectra of the digested PS showed radical oxidative reactions taking place in the mealworm body. The presence of hydroperoxide, alcohol and phenol groups was confirmed, and dimer fragments of styrene with quinone and phenol groups were obtained. The decrease in molecular weight and the alternation of benzene rings indicated that autoxidation and quinonization via phenolic intermediates occurred simultaneously in the mealworm body. The survival rate of mealworms reared on EPS was higher than that of starved worms, indicating that EPS was a nutrient source. However, no weight gain was observed in mealworms fed EPS alone. Comparison with the mixed diets with bran or urethane foams (PU) indicated that protein, phosphorus and magnesium components absent from EPS were required for mealworm growth.

## 1. Introduction

The amount of marine plastic litter is large and increasing [1] and has resulted in serious damage to the marine environment as microplastic (MP) pollution [2,3]. We are concerned that global pollution from MPs, smaller nanoplastics (NPs) and the additives contained in or adsorbed to them will develop into a “global toxic debt” as per Rilling et al. [4]. In particular, polystyrene (PS) are typical raw materials of MP, and the scale of their emissions is huge. For example, PS occupies 11% of the amount of resin production in Japan (in 2022). PS litter has resulted from the widespread commercial use of expanded PS (EPS) in building insulation and packaging such as cups and food trays, and has been fragmented by a combination of photo-oxidative degradation (autoxidation) caused by sunlight and friction caused by wind and waves to eventually exist in the oceans and atmosphere as MP. One way to prevent such MP conversions is to recycle PS. Recycling is preferable although there exist some problems such as biotic and chemical contaminations on the surface [5,6,7]. In particular, marine debris is colonized by microorganisms in the sea because a suitable substrate is photo-oxidatively produced on the surface under sunlight irradiation [1], and its surface adsorption of various organic compounds as well as biofilm formation occurs [8,9]. Removal of such contaminants during recycling requires thorough cleaning and sorting, which can significantly reduce the cost competitiveness of recycled products. In addition, PS recycling methods mainly involve chemical recycling at temperatures of 500 °C or higher [10,11], which requires expensive equipment and high operating costs. Mechanical recycling, which does not require high-temperature processes and can be performed with relatively inexpensive equipment, is a potential means of reducing the cost of plastic recycling compared to chemical recycling. However, the mechanical recycling process requires the shearing and cleaning of sunlight-degraded plastic waste in addition to the contamination mentioned above. The mechanical handling and cleaning can cause MP formation in plastic waste. In fact, Suzuki et al. recently studied plastic input and effluent volumes at three mechanical recycling facilities in Vietnam that handled electronics, bottles and household plastic waste and reported that during recycling, a large amount of MP was generated and released into the aquatic environment without proper treatment [12]. Brown et al. conducted a pilot study to investigate MP pollution from a mixed plastics recycling facility in the UK and also reported that the installation of filtration equipment was effective in reducing such pollution, demonstrating the need for additional equipment for mechanical recycling [13]. Furthermore, mechanical recycling has problems related to thermal oxidative degradation during the remolding process and the re-addition of antioxidants to prevent it [14]. In the molding process of PS and PP, which is carried out under high temperatures of around 200 °C, a decrease in molecular weight is certain to occur unless a phenolic primary antioxidant called AO is combined with a secondary antioxidant that is mainly composed of sulfur or phosphorus. It is necessary to add appropriate amounts of primary and secondary antioxidants when reprocessing waste plastics, but since it is difficult to determine the residual amounts of these antioxidants, additional additives may result in excessive amounts. The use of plastic products containing excessive amounts of additives can be harmful to health and should be avoided. The development of novel recycling methods that do not require precise sorting, expensive equipment, reactions at high temperatures and additional additives has been eagerly awaited.

Yang et al. confirmed that mealworms were able to digest EPS using various measurement methods, mainly spectroscopic instruments [15,16]. Many researchers are currently studying the biodegradation of various commodity plastics, primarily EPS and polyethylene (PE), by mealworms [17,18,19,20,21,22,23,24,25,26,27,28]. The ultimate goal of these studies is to remediate commodity plastics with mealworms. To date, it is clear that plastics such as EPS are oxidatively degraded by some types of bacteria in the gut [15,19,20,22,23,26,27]. Mealworms can digest not only PS and PE, but also various types of consumer plastics [21], and since they have digested leaves, branches and other wood waste under natural conditions, they can also digest plastic waste without having to separate it. Because mealworms are omnivorous, remediation can reduce the cost of plastic waste disposal, which is a significant economic benefit. Furthermore, if the plastic waste could be used to feed mealworms instead of remediation, the economic benefits would be immeasurable. Interestingly, a property of the gut microbiota changes when mealworms are grown on PS of different molecular weights [27], and feeding PS with a high concentration of the flame retardant hexabromocyclododecane (HBCD) leaves little residue in the mealworms’ bodies [29]. The digestion, i.e., biodegradation mechanism of PS by mealworms appears to be a radical oxidation reaction (autoxidation) using reactive oxygen species, similar to that of lignin by white rot fungi, because it occurs at a low temperature, such as in the body [30]. Furthermore, the molecular weight dependence of PS in the bacterial flora and the low persistence of cycloalkyl compounds such as HBCD in the body suggest that benzene and cycloalkyl ring degradation occurs in mealworms [31]. Cyclic compounds such as benzene rings are commonly used as antioxidants, UV absorbers and flame retardants in commodity plastics such as PS, and their residues in waste plastics are a problem in mechanical recycling, as mentioned above [14]. If these additives degrade with the plastic without remaining in the body, the mealworms themselves can be used as feed for livestock and farmed fish. This is a truly novel recycling method that should be called upcycling of commodity plastics such as PS by mealworms. There is no doubt that mealworms possess several enzymes that can initiate autoxidation reactions that cleave carbon–carbon bonds and ring-opening reactions that cleave benzene and cycloalkyl rings, but there are still no reported cases that have revealed the existence of these reactions. If it can be confirmed that mealworms carry out these reactions inside their bodies, it is expected that research to realize the upcycling of this novel waste plastic will be actively conducted around the world.

In this study, mealworms were reared on EPS alone, and PS in the discharged frass was analyzed by GPC, FT-IR and py-GC/MS. The changes in molecular weight and chemical structure before and after rearing were studied in detail to determine the biodegradation mechanism of PS by mealworms. In addition, differences in the growth behavior of mealworms on EPS alone, bran alone, EPS/bran and EPS/PU mixtures were studied, and elemental analysis was used to determine the nutrients required for weight gain in mealworms.

## 2. Materials and Methods

### 2.1. Materials

Mealworms (Tenebrio molitor) and bran (wheat bran) for feeding were purchased from Amazon.co.jp accessed on 16 June 2023. S-shaped 250 mm × 350 mm styrofoam (EPS) was commercially available with 90× foam magnification and was also purchased from Amazon.co.jp. The number-averaged molecular weight (Mn) was 8.6 × 10^4^, and the weight-averaged molecular weight (Mw) was 2.6 × 10^5^. The bran was used in its purchased particle form, with a grain size of approximately 1 mm. Urethane foams (PU) were purchased from Amazon.co.jp and then ground to granules of less than 1 mm using liquid nitrogen. This ground product served as the “PU” sample in the experiment. A PS sample for degraded PS preparation was purchased from Sigma-Aldrich Co. LLC (St Louis, MI, USA). The Mn and Mw were 1.7 × 10^5^ and 3.5 × 10^5^. Potassium persulfate (K_2_S_2_O_8_) and cuprous oxide were purchased from Wako Pure Chemical Industries (Osaka, Japan). Sea water was prepared using Gex artificial saltwater purchased from Amazon.co.jp. (Tokyo, Japan).

### 2.2. Mealworm Maintenance

Mealworms have feed left in their bodies immediately after the purchase, so they were fasted for two days to allow the feed in their bodies to be completely metabolized before being used for feeding (biodegradation) tests. 

### 2.3. Biodegradations of Expanded Polystyrene (EPS), EPS/Bran, Bran and EPS/Urethane Foam (PU)

All experiments were conducted using the 30 mealworms, which were selected to weigh a total of 3.53 ± 0.01 g. The four experimental setups (EPS, EPS/bran, bran and EPS/PU) were compared, along with a control experiment in which the mealworms were not fed. To assess the survival and degradation of PS, the 30 mealworms were transferred to a 120 cm^2^ cylindrical plastic container. These shells were removed once a day, and mealworms, diets and frass were weighed each week. In addition, any individuals that died or pupated during the rearing period were removed each week and a number of individuals equal to the average weight of the remaining individuals were added to maintain a total of 30 individuals. All experiments were repeated 4 times, and except for the control and the bran-only setups, two S-shaped pieces of EPS (with a total weight of 0.223 ± 0.003 g) were given. They were conducted in a thermostatic oven maintained at 20 °C. Mealworm mortality was assessed by counting dead mealworms every week over a total of four weeks, and dead larvae were then removed from the container. The EPS/bran mixture was placed directly into the cup with the weighed bran. At the weekly measurement, all residual bran in the cup was collected along with the frass, and a new amount of bran was added, equal to the amount given in the first week. The EPS/PU was also used as bait for mealworms using the same procedure as the EPS/bran. When only bran was fed, approximately 0.353 g (equivalent to one tenth of the total weight of 30 mealworms) of roughly 1 mm diameter powdered bran was given.

### 2.4. Method of Polystyrene (PS) Recovery from Frass

About 30 mg of the recovered frass was dissolved in 20 mL of THF for 1 h, and the solution was filtered through 1 µm filter paper. The total volume of the solution obtained was about 9 mL and this was transferred to a 9 mL screw tube. It was then concentrated in a stirrer at 50 °C for about half a day, and finally dried under reduced pressure at room temperature for half a day, and the PS residue remaining at the bottom of the screw tube was collected.

### 2.5. Degradation Using Sulfate Ion Radicals in Seawater (Enhanced Degradation Method)

A PS film was compress-molded into thin films (30 mm × 30 mm × 0.060 mm) by compression molding at 180 °C under 10 MPa for 11 min. The degradation was performed in seawater using sulfate ion radical. The procedure was according to our previous report [32]. (1) Several pieces of each film were put into a 100 mL glass vessel containing 20 mL of seawater solution with 0.54 g K_2_S_2_O_8_ at ca 65 °C for 12 h under stirring with a stirrer tip speed of ca 100 rpm. (2) The equal amount of K_2_S_2_O_8_ seawater solution was added to compensate for the consumption of oxidant, and its degradation was carried out for 12 h under the same conditions. (3) The five pieces of the film were then transferred to a new 100 mL glass vessel containing 20 mL of seawater solution with 0.54 g K_2_S_2_O_8_, and the degradations were started again under the same conditions. The enhanced degradation method was carried out for a predetermined number of 28 days using (1) to (3) as one set (total 28 sets). The pH value of the solution was changed from 8.2 to 3 during each set. The resulting sample was used as “degraded PS”.

### 2.6. Characterization and Analysis

The transform infrared spectra 16 scans were measured with a Fourier Transform Infrared spectrometer (FT-IR: Jasco FT-IR 660 plus) at a resolution of 4 cm^−1^ over the full mid-IR range (400–4000 cm^−1^). 

A multi-functional pyrolyzer (Frontier Labs. Fukusima Japan, EGA/PY-3030D) was attached to a gas chromatography/mass (GC/MS) spectroscopy (Py-GC/MS: SHIMADZU Kyoto Japan, GCMS-QP2010 PLUS). The measurement was employed with 100 μg sample. The pyrolysis was performed at 550 °C. Helium was used as the carrier gas for the capillary column with a flow rate of 1.0 mL/min. The MS system was operated under electron ionization mode at 70 eV.

The scanning electron microscope with energy dispersive X-ray spectroscopy (SEM/EDX) analysis was carried out with a JSM-7500FAM (JEOL, Tokyo Japan) at 5.0 kV. The working distance was about 3 × 4 mm. Samples were placed in a drying oven maintained at 27 °C for 30 min and were sputter-coated with gold before SEM imaging.

The gel permeation chromatography (GPC) sample in a small vial was dissolved in 5 mL of chloroform, and the obtained sample solution was directly measured with a SHIMADZU Prominence GPC system (SHIMAZU, Kyoto, Japan). The molecular weight was determined at 40 °C using chloroform as a solvent.

## 3. Results and Discussion

### 3.1. Biodegradation Mechanism of Polystyrene (PS)

To digest PS, the polymer chain must be definitely cleaved. Figure 1 depicts the molecular weight curves of PS samples. The weight average molecular weight (Mw) and molecular weight distribution (Mw/Mn) of the pristine PS (EPS) were 2.7 × 10^5^ and 3.2, respectively. On the other hand, the PS in the frass showed a 33% decrease in molecular weight, with Mw = 1.8 × 10^5^ and Mw/Mn = 2.9. The behavior reveals that the PS chain scission occurs in the mealworm’s body, most likely in the gut [16,17,20,26]. The changes in PS structure before and after the digestion, obtained from FT-IR spectra, provide powerful information about the chemical reactions taking place in the mealworm body. Furthermore, the PS structure can be compared to the PS structure altered by chemical oxidative degradation (autoxidation) to determine if the chemical reaction mechanism during digestion is the same or different. 

Comparisons of the pristine and degraded PS spectra with the digested PS spectrum are shown in Figure 2. In the higher wavenumber region of 4000–3000 cm^−1^, a relatively large peak around 3630 cm^−1^ of digested PS is observed, assigned to an isolated hydroperoxide, which has been reported in PP and PE [33]. The isolated hydroperoxide group peak is also present at relatively low intensity in the pristine PS (EPS) but disappears in the degraded PS. Instead, a broad peak assigned to an associated hydroperoxide group is present at 3550 cm^−1^ in the degraded PS. The associated hydroperoxide peak appears to be present in the digested PS, but overlaps with a very broad peak centered at 3250 cm^−1^ derived from alcohol and phenol groups, and its peak intensity is considerably lower than the isolated hydroperoxide group. In the 1900–1550 cm^−1^ region, which indicates oxidation (carbonylation) of the C-C main chain, the degraded PS shows a ketone group peak centered at about 1700 cm^−1^. On the other hand, in the digested PS, the corresponding peak shifts to the low wavenumber side (about 1670 cm^−1^) and is observed as an α,β-unsaturated ketone group. A peak assigned to benzyl alcohol is observed at 1430 cm^−1^ only in the digested PS in the wavelength range around the phenyl group (1550–1350 cm^−1^). If only autoxidation occurs, it would not explain the presence of α,β-unsaturated ketone and benzyl alcohol groups observed in the IR peaks of the digested PS. The IR spectra results suggest that not only autoxidation but also other reactions are involved in the mechanism of PS biodegradation by mealworms. Peaks not present in PS were observed in the py-GC/MS spectrum of digested PS. As shown in Figure 3, the mass number *m*/*z* = 251 observed in the digested PS was attributed to the chemical structure of a carbonyl terminus attached to a styrene dimer from the low mass number fragments observed. In addition, fine fragments ranging in mass number from *m*/*z* = 85 to 45 suggest the presence of a 5-carbon carbonyl group. Based on the results of the fragmentation analysis described above, we expect to find a dimer of styrene with a carbonyl terminus having a five-carbon chain at the end. However, the carbonyl moiety, which has a five-carbon chain, is not very thermally stable. In its directly bound form in the styrene dimer, it would be difficult to detect directly by py-GC/MS. The 5-carbon carbonyl group is probably formed by the opening of the benzene ring via quinination. The quinone precursors appear to be phenolic compounds as observed by FT-IR measurement. The presence of thermally relatively stable phenolic compounds in digested PS can also be confirmed by py-GC/MS measurements. Attempts were made to confirm the formation of phenolic compounds, and the py-GC/MS spectrum at 24.24 min showed peaks attributed to phenol (*m*/*z* = 95, 110, 127, 222, 267) and benzylphenol (*m*/*z* = 143, 282) compounds (see Figure 4). These peaks would be derived from dimer fragments of styrene with quinone and phenol groups, as shown in Figure 4. It suggests that the benzene ring is decomposed by quinonization via a phenolic intermediate.

The decrease in PS molecular weight after digestion, the presence of oxidation-derived functional groups, and the phenolic and quinone compounds produced are useful in estimating the mechanism of PS degradation that occurs in the mealworm body. The decrease in molecular weight means that autoxidation is certain to occur and can trigger polymer chain scission even at relatively low temperatures, such as in the body of mealworms. Figure 5 shows the mechanism of PS chain scission by autoxidation. The hydroperoxide peak clearly observed in the IR spectrum of digested PS is evidence that autoxidation occurs. Since most of the hydroperoxides in digested PS are of the isolated type as mentioned above, the formation of a polymer chain end with an acetophenone structure (*m*/*z* = 120) is expected first (Figure 5). However, no peak with the mass *m*/*z* = 120 is observed in the MS spectra of Figure 3 and Figure 4. The absence of the acetophenone fragment indicates that another substance with a modified benzene ring is simultaneously produced by another reaction. A styrene dimer (*m*/*z* = 251) with a terminal carbonyl group is observed (Figure 3 and Figure 4). At the same time, numerous peaks are observed in a wide range of *m*/*z* = 45–85, suggesting the presence of a pentacarbonyl compound. The benzene ring would be opened via 2-acetophenol, eventually producing a dicarbonyl compound with a terminal carboxylic acid as shown in Figure 3 and Figure 4 [31]. The IR spectrum supports the presence of a compound with a 2-acetophenol structure, as a phenol group peak is observed around 3250 cm^−1^ in the digested PS. The main reaction route of PS chain scission by autoxidation would be the 2-acetophenol end producing route, not the acetophenone end (Figure 5). The dicarbonyl compound with a terminal carboxylic acid is formed by the same mechanism as a photo-oxidation reaction. That is, it is a yellowing reaction of PS caused by the ring-opening reaction of the benzene ring [31]. Figure 6 depicts the ring-opening reaction mechanism of benzene rings. The peak attributed to benzyl alcohol is shown in the IR chart of the digested PS. In addition, the MS chart in Figure 4 shows fragments at *m*/*z* = 282 and 143 that can be attributed to the benzylphenol structure. In light of these results, the dicarbonyl structure may have two possible pathways of formation, at the polymer chain end and in the polymer chain, respectively. Both pathways, as shown in Figure 6, open the ring via quinone-like structures and peroxide radicals due to oxidation of phenol to form dicarbonyl compounds. In Figure 5, the mechanism of PS biodegradation by mealworms is not only autoxidation, which is the polymer chain scission reaction, but also a process that proceeds to other reaction mechanisms. Furthermore, Figure 6 shows that the reaction mechanism is the ring-opening reaction of the benzene ring of the side chain.

Figure 7 shows the schematic diagram of PS biodegradation by mealworms. When mealworms feed on the EPS, enzymes in the gut generate reactive oxygen species that cause autoxidation, which cleaves the polymer chains. During or after the chain scission, the benzene ring is converted to an unstable quinone compound via a phenolic compound using a similar active oxygen (probably OH radical). This final quinone compound is converted to an α,β-unsaturated ketone (α,β-unsaturated carboxylic acid) that can be decomposed and metabolized. The results of FT-IR and py-GC/MS measurements of digested PS suggest the biodegradation mechanism shown in Figure 7.

### 3.2. Effect of Bran Addition on Mealworm Growth

Figure 8 depicts the mortality and weight change (*w*/*w*_0_) of starved and EPS-raised mealworms during each rearing period. The starved mealworms had zero mortality after one week, but mortality increased rapidly with rearing, rising to ca 20% after 4 weeks. On the other hand, the mortality rate of mealworms reared in the EPS remained at 2% until the first 2 weeks of rearing, then increased slightly to 5% at 3 weeks and 7% at 4 weeks. Clearly, the higher survival rates are for mealworms that have ingested EPS, and it is clear that EPS is a nutrient source [15,16]. However, no weight gain was observed in the mealworms that ingested only EPS, and their body weight was reduced to 87% of the original weight after 4 weeks of rearing (Figure 8). In addition, the starved mealworms weighed an additional 3% less than those that ingested EPS up to 3 weeks of rearing, after which they began cannibalism, which was not seen in the EPS ones. These results suggest that EPS is a nutrient source for mealworms and that some nutrients are also lacking. Rumbos et al. reported that bran added to plastics such as PS facilitated the growth of mealworms [19]. It appears that the nutrients lacking in EPS are present in bran. Appendix A shows the dependence of mealworm weight (total 30 worms) on rearing period in EPS and bran-mixed diets with different mixing ratios. The mealworms reared on the diets with a bran mixture ratio of 10:2 or higher showed gradual weight gain up to 4 weeks of rearing. EPS is an alternative nutrient to carbohydrates because it is composed only of C and H elements, but it cannot be a protein source because it does not contain N. Since bran is rich in protein, its addition compensates for the deficient protein. Other important nutrients may include phosphorus and minerals. Phosphorus, in particular, is an essential element that builds the genes of the organism and is an essential nutrient for the growth of mealworms. Appendix A shows the SEM/EDX analysis of frass passed from the body of a mealworm fed only bran as feed after 1 week of rearing. In fact, the undigested bran was found to contain a 2.26% phosphorus component. An interesting point is the presence of the element magnesium although its content is low at 1%. The deficiency induces oxidative stress in an organism cell, such as in a mulberry plant [34]. Magnesium compounds are known to act as a coenzyme for superoxide dismutase (SOD), which protects cells from such oxidative stress [35]. Since the decomposition mechanism of EPS utilizes active oxygen species such as OH radicals, which are a source of oxidative stress, a cytoprotective enzyme such as SOD is also required. EPS alone is not sufficient for rearing mealworms and feeds such as bran containing protein; phosphorus and magnesium compounds are required as supplements. In fact, when the mealworms were fed a mixed diet with polyurethane (PU) powder added to EPS instead of bran, their body weight decreased over the rearing period, as shown in Appendix A, at about the same rate of decrease as when fed only EPS. PU is considered to be an N source because it has urethane bonds. However, this result suggests that the ingestion of PU does not substitute for protein. Protein is an essential material for making various enzymes. Mealworms can digest PU as N source, but cannot convert them to protein in their bodies. In addition, it does not contain phosphorus or mineral components. The use of mealworms as a method to clean up plastic waste requires research into the optimal ratio of these supplemental nutrients. More research is needed. 

## 4. Conclusions

A 33% decrease in molecular weight was observed for PS in the frass after 1 week of feeding to mealworms. The result revealed that the PS chain scission occurred in the mealworm’s body. The large isolated hydroperoxide peak was observed around 3630 cm^−1^ in the digested PS by the FT-IR measurement. On the other hand, it disappeared in the degraded PS. Instead, the associated hydroperoxide peak was present at 3550 cm^−1^. The associated hydroperoxide peak appeared to be present in the digested PS, but overlapped with a very broad peak at around 3250 cm^−1^ derived from alcohol and phenol groups. The peak intensity was considerably lower than the isolated hydroperoxide group. In addition, the FT IR peaks derived from α,β-unsaturated ketone and benzyl alcohol groups were observed at 1670 and 1430 cm^−1^, respectively, in the digested PS. The phenol and alcohol groups observed in the FT-IR measurements were determined by py-GC/MS analysis to be derived from phenol and benzyl alcohol compounds. The presence of α,β-unsaturated ketone was also confirmed by the py-GC/MS measurements. These peaks were derived from dimer fragments of styrene with quinone and phenol groups, indicating that the benzene ring was decomposed by quinonization via a phenolic intermediate. This was the same mechanism as a yellowing reaction of PS caused by the ring-opening reaction of the benzene ring. The decrease in molecular weight showed that autoxidation occurred in the body of the mealworm, and the FT-IR and py-GC/MS spectra indicated that the α,β-unsaturated ketone compound using the yellowing reaction was produced simultaneously. The survival rate of mealworms reared in EPS was higher than that of starved worms, showing that EPS was a nutrient source. However, no weight gain was observed in the mealworms that ingested only EPS. Similar weight gain was observed to the mixed diet of EPS and bran. The results suggested that EPS was a nutrient source for mealworms, but that some nutrients were missing. Protein was considered the main deficient nutrient because it was a nutrient found in bran but not in EPS. In addition, the SEM/EDX analysis of the frass from the bodies of the mealworms indicated that phosphorus and magnesium compounds also appeared to be deficient nutrients. The mixed diet with PU added to EPS as an N source was fed to mealworms, but no weight gain was observed, in contrast to the bran mixture. The upcycling of EPS waste with mealworms would require the addition of natural protein and mineral components such as phosphorus and magnesium.

## Figures and Tables

**Figure 1 polymers-16-01632-f001:**
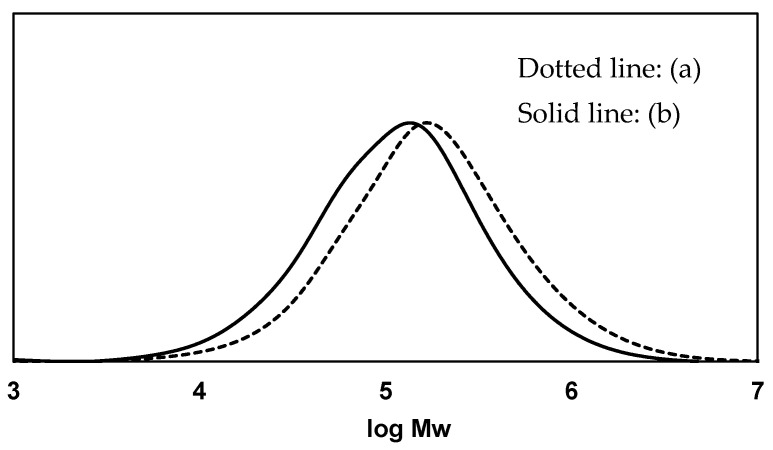
Molecular weight curves of PS samples: (a) Pristine PS (EPS). (b) PS in frass after 1 week of feeding.

**Figure 2 polymers-16-01632-f002:**
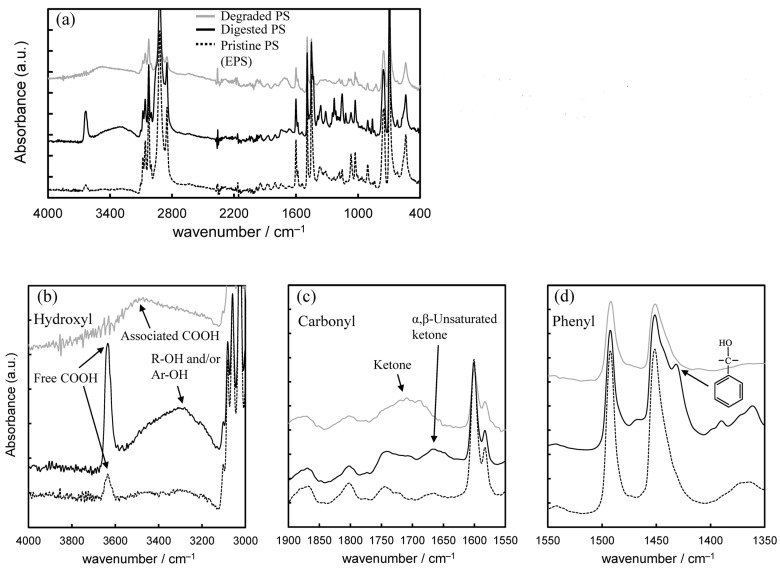
Comparisons of FT-IR spectra for all measurement, hydroxyl, carbonyl and phenyl absorption regions: (**a**) All, (**b**) Hydroxyl, (**c**) Carbonyl, (**d**) Phenyl.

**Figure 3 polymers-16-01632-f003:**
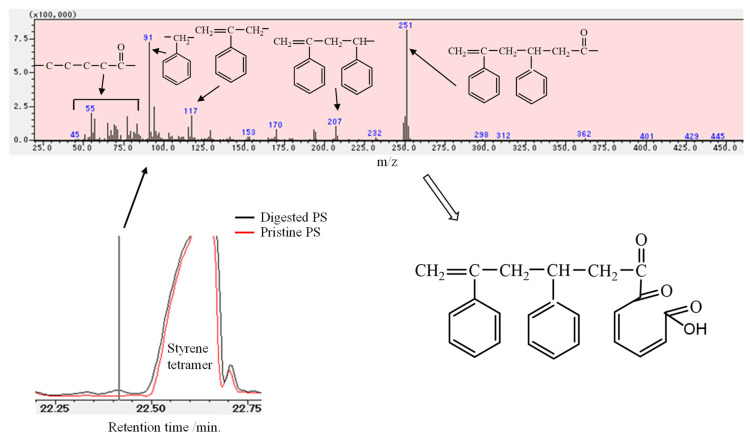
Py-GC/MS spectrum just before styrene tetramer (at 22.41 min).

**Figure 4 polymers-16-01632-f004:**
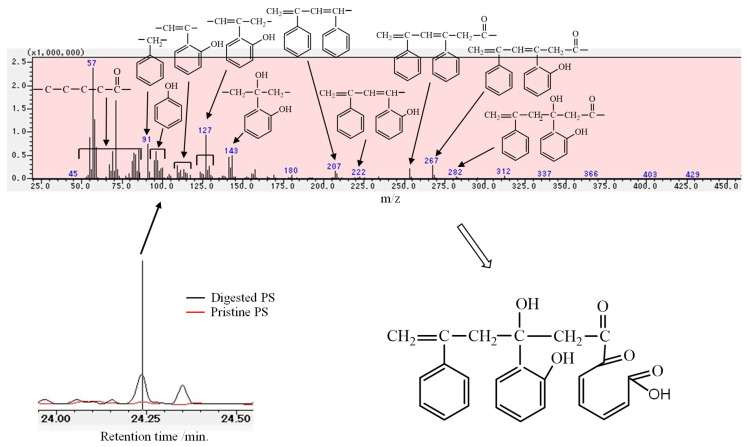
Py-GC/MS spectrum at 24.24 min.

**Figure 5 polymers-16-01632-f005:**
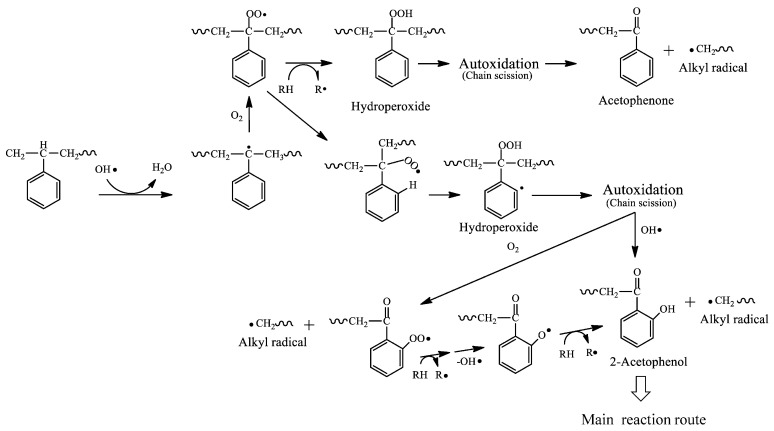
Scission mechanism of polystyrene main chain by autoxidation and chain end structures.

**Figure 6 polymers-16-01632-f006:**
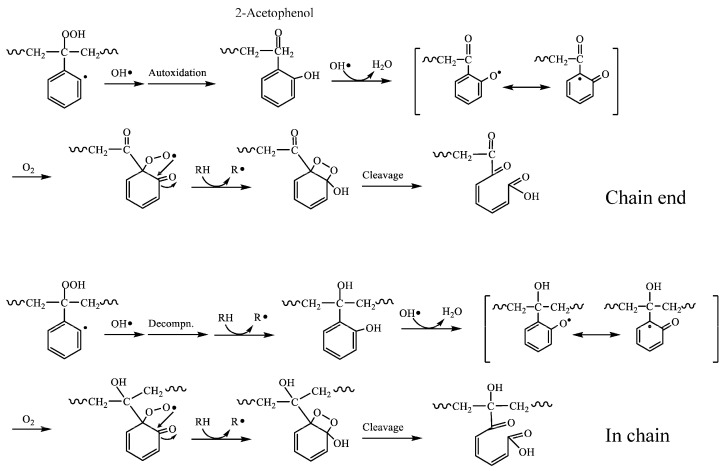
Ring-opening reaction mechanism of benzene rings at chain end and in chain.

**Figure 7 polymers-16-01632-f007:**
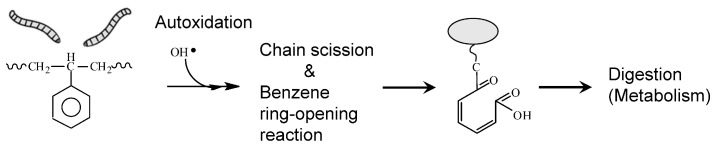
Schematic diagram of biodegradation mechanism.

**Figure 8 polymers-16-01632-f008:**
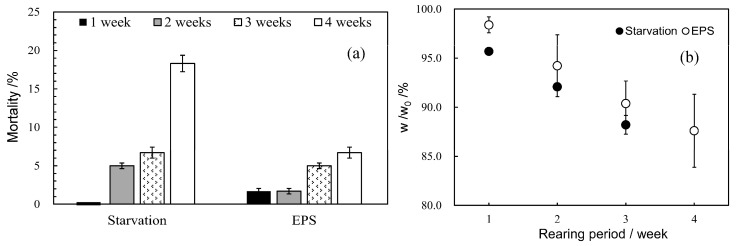
Mortality and weight change (*w*/*w*_0_) of starved and EPS-raised mealworms during each rearing period: (**a**) Mortality, (**b**) Weight change (*w*/*w*_0_).

## Data Availability

Data is contained within the article.

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
