# Peer review of "Biodegradation Mechanism of Polystyrene by Mealworms (Tenebrio molitor) and Nutrients Influencing Their Growth"

_polymers, 2024, doi:10.3390/polym16121632_

Round 1

Reviewer 1 Report

Comments and Suggestions for Authors

Title: Biodegradation mechanism of polystyrene by mealworms (Tenebrio molitor) and nutrients influencing their growth.

1. Avoid more citation in the single line text. In this manuscript the authors cited like this ...........the amount of marine plastic litter is large and increasing [1–5]. This has to be avoided. Analyze each cited reference thoroughly and give authors own inference.

2. Section 2.4 Method of PS recovery from Frass need to be explained more for readers clarity.

3. Avoid abbreviation in section headings.

4. Quality of FT-IR spectra presented in fig. 2 to be improved. More interpretation of these graphs is required.

5. There is no much difference between mechanisms presented in Fig. 5 and Fig. 6. Justification required.

6. Conclusion must be precise. Include major numerical findings in the conclusion. 

7. Language editing of the manuscript is required. 

Comments on the Quality of English Language

Moderate editing of English language required

Author Response

Reviewer 1

Title: Biodegradation mechanism of polystyrene by mealworms (Tenebrio molitor) and nutrients influencing their growth.

Comment 1. Avoid more citation in the single line text. In this manuscript the authors cited like this ...........the amount of marine plastic litter is large and increasing [1–5]. This has to be avoided. Analyze each cited reference thoroughly and give authors own inference.

Answer: We have revised it accordingly.

Comment 2. Section 2.4 Method of PS recovery from Frass need to be explained more for readers clarity.

Answer: We revised it as follows: About 30 mg of the recovered frass was dissolved in 20 mL of THF for 1 hour, and the solution was filtered through 1 µm filter paper. The total volume of the solution obtained was about 9 mL and transferred to a 9 mL screw tube. It was then concentrated in a stirrer at 50 °C for about half a day, and finally dried under reduced pressure at room temperature for half a day, and the PS residue remaining at the bottom of the screw tube was collected.

Comment 3. Avoid abbreviation in section headings.

Answer: We revised them.

Comment 4. Quality of FT-IR spectra presented in fig. 2 to be improved. More interpretation of these graphs is required.

Answer: We improved Figure 2 and add the sentences as follows: If only autoxidation occurs, it would not explain the presence of α, β-unsaturated ke-tone and benzyl alcohol groups observed in the IR peaks of the digested PS. The IR spectra results suggest that not only autoxidation but also other reactions are involved in the mechanism of PS biodegradation by mealworms.

Comment 5. There is no much difference between mechanisms presented in Fig. 5 and Fig. 6. Justification required.

Answer: We added sentences to explain the difference as follows: In Figure 5, the mechanism of PS biodegradation by mealworms is not only autoxidation, which is the polymer chain scission reaction, but also a process that proceeds to other reaction mechanisms. Furthermore, Figure 6 shows that the reaction mechanism is the ring-opening reaction of the benzene ring of the side chain.

Comment 6. Conclusion must be precise. Include major numerical findings in the conclusion.

Answer: We revised our text.

Comment 7. Language editing of the manuscript is required.

Answer: We revised our text.

Reviewer 2 Report

Comments and Suggestions for Authors

The manuscript is subjected to some improvement before being accepted for publication:

  1. Line # 132, The experimental setup representation should be provided for better understanding.
  2. Many references used in the study are old one, author should use recent appropriate references.
  3. The schematic diagram of biodegradation mechanism is missing.
  4. Line 177-196, The sections 2.6-2.9 should be merged in single section of characterization and analysis.
  5. Figure sub-sections (a, b, c) are not mentioned in figures and captions such as in Figure 2, 7 and others.
  6. The conclusion section should be rewritten with important findings.
  7.  

Author Response

Reviewer 2

The manuscript is subjected to some improvement before being accepted for publication:

  1. Line # 132, The experimental setup representation should be provided for better understanding.

Answer: We revised it as follows: Mealworms have feed left in their bodies immediately after the purchase, so they were fasted for two days to allow the feed in their bodies to be completely metabolized before being used for feeding (biodegradation) tests.

  1. Many references used in the study are old one, author should use recent appropriate references.

Answer: Microplastics research references have been scraped from the old and new ones added (Ref. 1-7). However, Revised Refs. 30, 31, 33, 34, and 35 have not been revised. The studies of ligninolytic enzymes, polystyrene and polyethylene degradation and antioxidant enzymes are older, and these references contain important findings and are highly original. We do not believe it is appropriate to replace these references with newer ones.

  1. The schematic diagram of biodegradation mechanism is missing.

Answer: We added the schematic diagram of biodegradation mechanism as new Figure 7. In addition, we add the sentences as follows: Figure 7 shows the schematic diagram of PS biodegradation by mealworms. When mealworms feed on the EPS, enzymes in the gut generate reactive oxygen species that cause autoxidation, which cleaves the polymer chains. During or after the chain scis-sion, the benzene ring is converted to an unstable quinone compound via a phenolic compound using a similar active oxygen (probably OH radical). This final quinone compound is converted to an α,β-unsaturated ketone (α,β-unsaturated carboxylic ac-id) that can be decomposed and metabolized. The results of FT-IR and py-GC/MS measurements of digested PS suggest the biodegradation mechanism shown in Figure 7.

  1. Line 177-196, The sections 2.6-2.9 should be merged in single section of characterization and analysis.

Answer: We merged them.

  1. Figure sub-sections (a, b, c) are not mentioned in figures and captions such as in Figure 2, 7 and others.

Answer: We revised them in Figure 2 & new Figure 8.

  1. The conclusion section should be rewritten with important findings.

Answer: We revised our text.

Round 2

Reviewer 1 Report

Comments and Suggestions for Authors

Title: Biodegradation mechanism of polystyrene by mealworms (Tenebrio molitor) and nutrients influencing their growth.

The revised manuscript is improved and the authors are responded well for the queries raised by the Reviewers. Now I recommend this manuscript for publication in this Journal.

Comments on the Quality of English Language

Minor editing of English language required

Reviewer 2 Report

Comments and Suggestions for Authors

The authors have revised accordingly.